# Association of Infection with Different SARS-CoV-2 Variants during Pregnancy with Maternal and Perinatal Outcomes: A Systematic Review and Meta-Analysis

**DOI:** 10.3390/ijerph192315932

**Published:** 2022-11-29

**Authors:** Jie Deng, Yirui Ma, Qiao Liu, Min Du, Min Liu, Jue Liu

**Affiliations:** 1Department of Epidemiology and Biostatistics, School of Public Health, Peking University, Address No. 38, Xueyuan Road, Haidian District, Beijing 100191, China; 2Institute for Global Health and Development, Peking University, Address No. 5, Yiheyuan Road, Haidian District, Beijing 100871, China

**Keywords:** COVID-19, SARS-CoV-2, pregnancy, variant, maternal outcome, perinatal outcome, systematic review, meta-analysis

## Abstract

The aim of this study is to review the currently available data, and to explore the association of infection with different severe acute respiratory syndrome coronavirus 2 (SARS-CoV-2) variants during pregnancy with maternal and perinatal outcomes in the real world. Observational cohort studies were analyzed that described the maternal and perinatal outcomes of infection with different SARS-CoV-2 variants during pregnancy. Random-effects inverse-variance models were used to evaluate the pooled prevalence (PP) and its 95% confidence interval (CI) for maternal and perinatal outcomes. Random effects were used to estimate the pooled odds ratios (OR) and their 95% CI for different outcomes between Delta and pre-Delta periods, and between Omicron and Delta periods. Eighteen studies, involving a total of 133,058 cases of SARS-CoV-2 infection during pregnancy (99,567 cases of SARS-CoV-2 wild type or pre-variant infection and 33,494 cases of SARS-CoV-2 variant infections), were included in this meta-analysis. Among pregnant women with SARS-CoV-2 infections, the PPs for required respiratory support, severe or critical illness, intensive care unit (ICU) admission, maternal death, and preterm birth <37 weeks were, respectively, 27.24% (95%CI, 20.51–33.97%), 24.96% (95%CI, 15.96–33.96%), 11.31% (95%CI, 4.00–18.61%), 4.20% (95%CI, 1.43–6.97%), and 33.85% (95%CI, 21.54–46.17%) in the Delta period, which were higher than those in the pre-Delta period, while the corresponding PPs were, respectively, 10.74% (95%CI, 6.05–15.46%), 11.99% (95%CI, 6.23–17.74%), 4.17% (95%CI, 1.53–6.80%), 0.63% (95%CI, 0.05–1.20%), and 18.58% (95%CI, 9.52–27.65%). The PPs for required respiratory support, severe or critical illness, and ICU admission were, respectively, 2.63% (95%CI, 0.98–4.28%), 1.11% (95%CI, 0.29–1.94%), and 1.83% (95%CI, 0.85–2.81%) in the Omicron period, which were lower than those in the pre-Delta and Delta periods. These results suggest that Omicron infections are associated with less severe maternal and neonatal adverse outcomes, though maternal ICU admission, the need for respiratory support, and preterm birth did also occur with Omicron infections. Since Omicron is currently the predominant strain globally, and has the highest rates of transmission, it is still important to remain vigilant in protecting the vulnerable populations of mothers and infants. In particular, obstetricians and gynecologists should not ignore the adverse risks of maternal ICU admission, respiratory support, and preterm births in pregnant patients with SARS-CoV-2 infections, in order to protect the health of mothers and infants.

## 1. Introduction

Caused by severe acute respiratory syndrome coronavirus 2 (SARS-CoV-2), the 2019 coronavirus pandemic (COVID-19) has posed an extraordinary threat and a huge burden to global public health and economic systems [1]. With the increased numbers and persistent changes in the virus, variants of concern (VOC) such as Alpha, Beta, Gamma, Delta, and Omicron have emerged, with the Omicron variant being the one currently circulating most rapidly [2]. Globally, as of 26 September 2022, it was incompletely reported that there have been more than 612 million confirmed cases of COVID-19, including more than 6 million deaths [3]. Pregnant women also accounted for a certain proportion of these confirmed infections and confirmed deaths.

Pregnant women are at high risk of contracting COVID-19, and the impact of infection with SARS-CoV-2 during pregnancy on mothers and their unborn babies has been a topic of great concern in related research fields throughout the world [4,5]. Previous studies have shown that pregnant women are at increased risk of severe illness associated with COVID-19, and that COVID-19 during pregnancy is associated with an increased risk of adverse pregnancy outcomes, as well as maternal and neonatal complications [6,7,8]. With the rise of different SARS-CoV-2 variants, the outcomes of maternal infection with these different variants has attracted wider interest among researchers. Data from the United Kingdom (UK) obstetric surveillance system national cohort suggested that, compared to the wild type period, symptomatic women admitted in the Alpha period were more likely to require respiratory support, have pneumonia, and be admitted to intensive care [9]. In addition, women admitted during the Delta period had a further increased risk of pneumonia [9]. Some existing studies showed that compared with previous VOCs such as B.1.1.7 (Alpha), B.1.351 (Beta), and P.1 (Gamma), disease severity in pregnant women infected with SARS-CoV-2 was worse during the predominance of the B.1.617.2 (Delta) strain [10,11]. However, subsequent statistics suggested that infection with the highly contagious B.1.1.529 (Omicron) variant during pregnancy was associated with milder disease outcomes compared to the preceding Delta variant [12,13,14], which was likely related to higher vaccination coverage, lower virulence of the Omicron variant, and infection-acquired immunity [13].

A living systematic review and meta-analysis carried out by John Allotey et al. suggested that the risk of stillbirth (OR = 1.81, 95%CI, 1.38–2.37) and admission to the neonatal intensive care unit (NICU, OR = 2.18, 95%CI, 1.46–3.26) were higher in babies born to women with COVID-19 compared with those without COVID-19 [6]. In a national cohort in the UK, while the majority of babies were born live, babies born to mothers in the Alpha period were more likely to require admission to hospital for neonatal care than in the wild type period [9]. In addition, reports of SARS-CoV-2 placentitis were rare in the first wave of outbreaks caused by the original wild type but became increasingly common in the pandemic waves caused by the Alpha and Delta variants [15,16], while SARS-CoV-2 placentitis might be associated with fetal hypoxic–ischemic injury resulting from severe placental damage but not necessarily equivalent to fetal infection [8]. Data from a national maternal surveillance platform in Malawi suggested that the rates of neonatal stillbirths or death were significantly lower in the Omicron-dominated fourth wave than in the Beta- and Delta-dominated second and third waves combined [14].

Pregnant women undergo a series of physiological changes in various bodily systems during pregnancy. As a high-risk group within the population, they not only need to meet the challenges and uncertainties related to pregnancy status but also the challenges and uncertainties related to SARS-CoV-2, especially those involving different variants [5]. The Secretary of Primary Health Care at the Brazilian health ministry once reportedly asked Brazilian women ‘if possible’ to delay pregnancy over COVID-19 variants [17]. Different SARS-CoV-2 variants might have different impacts on pregnant women and their fetuses. Some of the variants might also cause more serious consequences, while some of them may be less serious but might still bring significant pressure to the medical care system. Therefore, understanding the effects of infection with different SARS-CoV-2 variants during pregnancy on maternal and perinatal outcomes can play an important role in maternal health care and the rational allocation of medical resources. Although the impact of different SARS-CoV-2 variants is an area of global concern, there are very few published studies exploring the impact of different SARS-CoV-2 variants on pregnancy and perinatal outcomes. Thus, in this systematic review and meta-analysis, the aim is to review the available data to explore the association between different SARS-CoV-2 variant infections during pregnancy with maternal and perinatal outcomes in the real world.

## 2. Methods

### 2.1. Search Strategy and Selection Criteria

The data analyzed for this study included papers published up until 14 August 2022, without language restrictions, on PubMed, EMBASE, Web of Science, Science Direct, medRxiv, and bioRxiv databases with the following search terms: (SARS-CoV-2 OR COVID-19 OR coronavirus) AND (variant OR VOC OR Alpha OR Beta OR Gamma OR Delta OR Omicron OR B.1.1.529 OR B.1.1.7 OR B.1.351 OR P.1 OR B.1.617.2) AND (gestation OR gravidity OR pregnancy OR fetation OR conception).

EndNoteX8.2 (Thomson Research Soft, Stanford, CA, USA) was used to manage the records. This study was performed in strict accordance with the Preferred Reporting Items for Systematic Reviews and Meta-Analyses (PRISMA in the Appendix A) [18]. This study was registered on PROSPERO (CRD42022342089).

Studies were included that examined the association of infection with different SARS-CoV-2 variants during pregnancy with maternal and perinatal outcomes. Inclusion criteria were: (1) studies that explored maternal and perinatal outcomes of SARS-CoV-2 infection during pregnancy, (2) specified the type of maternal SARS-CoV-2 variant infection or the dominant epidemic strains during the infection, and (3) published studies or preprinted studies. The following studies were excluded: (1) studies irrelevant to the subject of the meta-analysis, such as studies that did not use SARS-CoV-2 infections during pregnancy as the exposure; (2) those with insufficient data to calculate the rate of maternal and perinatal outcomes; (3) duplicate studies or overlapping participants; (4) reviews, editorials, conference papers, case reports, or animal experiments; and (5) studies that did not clarify the identification of COVID-19. For example, the confirmed diagnosis of COVID-19 via reverse-transcription polymerase chain reaction (rt-PCR) test, serologic test, or other means was not mentioned in the text.

Studies were identified by two investigators (DJ and MYR) independently following the criteria above, while discrepancies were solved by means of consensus or by a third investigator (LQ).

### 2.2. Data Extraction

The following data were extracted: (1) basic information about the studies, including the first author, publication time, article type (preprinted or published article), study design, and location where the study was conducted; (2) characteristics of the study population, including population sizes, age, gestational age at diagnosis, vaccination status, and type of variant; (3) number or proportion of cases who had maternal outcomes, pregnancy outcomes, perinatal outcomes, pregnancy outcomes; and (4) number or proportion of cases with different severities of disease, form of respiratory support, form of delivery, and drug therapy.

Data extraction was independently conducted by two investigators (DJ and MYR) following the criteria above, while discrepancies were solved by consensus or by a third investigator (LQ).

### 2.3. Quality Assessment and Publication Bias

The Newcastle–Ottawa quality assessment scale was used to evaluate the risk of bias in cohort studies. Cohort studies were classified as having a low (≥7 stars), moderate (5–6 stars), and high risk of bias (≤4 stars), with an overall quality score of 9 stars. Quality assessment was conducted independently by two investigators (DJ and MYR), while discrepancies were solved by consensus or by a third investigator (LQ). Publication bias was evaluated for the comparison of maternal and perinatal outcomes between different SARS-CoV-2 variants based on a modified Harbord’s test, and a *p*-value >0.1 indicated no publication bias.

### 2.4. Data Synthesis and Statistical Analysis

A meta-analysis was carried out to estimate the pooled prevalence (PP) and its 95% confidence interval (CI) for maternal and perinatal outcomes after infection with different variants during pregnancy. The odds ratio (OR) and its 95% CI was also estimated for different outcomes between Delta and pre-Delta periods, and between Omicron and Delta periods. Random-effects or fixed-effects models were used to separately pool the rates and the adjusted estimates across studies, based on the heterogeneity between estimates (I^2^). Fixed-effects models were used if I^2^ ≤ 50%, which represented low to moderate heterogeneity, while random-effects models were used if I^2^ ≥ 50%, representing substantial heterogeneity. The D-L method was used to estimate the tau square in the case of a random-effects model. Publication bias was assessed by means of Harbord’s modified test. All analyses used Stata version 16.0 (Stata Corp, College Station, TX, USA).

## 3. Results

### 3.1. Basic Characteristics

As part of the initial literature research, 1066 potential articles were identified up to 14 August 2022, including published articles (184 in PubMed, 160 in Embase, 483 in Web of Science, 76 in ScienceDirect) and 163 preprinted articles (43 in bioRxiv, 120 in medRxiv), and 373 duplicates were excluded. After reading the titles and abstracts, 650 articles were excluded based on the inclusion and exclusion criteria. Among the 43 studies under full-text review, 25 studies were excluded. Eventually, 18 studies (including 17 published articles and 1 preprinted article) were included in this meta-analysis based on the inclusion criteria. The literature retrieval flow chart is shown in Figure 1**.**

Of the 18 studies included [9,10,12,19,20,21,22,23,24,25,26,27,28,29,30,31,32,33], most were limited to America (*n* = 6), followed by Turkey and the UK (3 in Turkey, 2 in Turkey and the UK, 1 in the UK). The included studies described maternal and perinatal outcomes of different SARS-CoV-2 variant infections during pregnancy, involving a total of 133,058 cases of SARS-CoV-2 infection during pregnancy, and 99,567 cases of SARS-CoV-2 wild type or pre-variant infections (4129 wild type infections, 93,825 pre-Delta infections, 1416 pre-variant infections, 130 pre-Omicron infections, 67 pre-Alpha infections) and 33,494 cases of SARS-CoV-2 variant infections (2587 Alpha infections, 29,016 Delta infections, 1843 Omicron infections, 25 Alpha or Gamma infections, 23 Alpha or Beta infections). The characteristics of the included studies are shown in Appendix A**.**

### 3.2. Pooled Prevalence (PP) of Maternal and Perinatal Outcomes of Different SARS-CoV-2 Variant Infections during Pregnancy

The PP of maternal and perinatal outcomes of different SARS-CoV-2 variant infections during pregnancy, as shown in Table 1.

#### 3.2.1. Wild Type SARS-CoV-2 Infection during Pregnancy

Among pregnant women infected with the wild type of SARS-CoV-2, the PP for maternal death was only 0.39% (95%CI, −0.44–1.22%). Pneumonia (PP = 18.95%, 95%CI, 17.01%-20.90%) and gestational hypertension (PP = 6.62%, 95%CI, −4.84–18.08%) were common pregnancy complications. In total, 13.4% (95%CI, 10.55–16.25%) of cases needed respiratory support, and 0.16% (95%CI, −0.74–1.06%) needed extracorporeal membrane oxygenation (ECMO), which were lower than for those infected with other variants. The PP of live birth, spontaneous vaginal delivery, cesarean section, and abortion were 81.7% (95%CI, 44.94–118.45%), 47% (95%CI, 27.20–66.80%), 41.7% (95%CI, 31.40–52.00%), and 2.65% (95%CI, 2.05–3.26%), respectively. In addition, the PP of adverse perinatal outcomes such as stillbirth (PP = 0.89%, 95%CI, 0.56–1.22%), neonatal death (PP = 0.29%, 95%CI, −0.04–0.62%), and NICU admission (PP = 12.2%, 95%CI, 7.98–16.41%) were lower than those infected with other variants.

#### 3.2.2. SARS-CoV-2 Alpha or Gamma Variant Infections during Pregnancy

The severity of disease in pregnant women infected with the Alpha variant was less severe, with only 0.28% (95%CI, 0.04–0.53%) of maternal deaths, 4.21% (95%CI, 2.82–5.61%) requiring invasive ventilation or ECMO, and 0.54% (95%CI, 0.03–1.06%) requiring ECMO. The PPs for spontaneous vaginal delivery and cesarean section were 50.18% (95%CI, 20.16–80.19%) and 37.32% (95%CI, 24.58–50.05%), respectively. In addition, the PP for preterm births (<37 weeks) (PP = 36.12%, 95%CI, 5.37–66.87%) was higher, while the PP for stillbirths (PP = 0.77%, 95%CI, 0.16–1.38%) was lower than those infected with other variants.

Among pregnant women infected with the Alpha and Gamma variants, 7.35% (95%CI, 1.15–13.56%) required invasive ventilation or ECMO and 7.35% (95%CI, 1.15–13.56%) required ECMO, which were higher than for those infected with other variants. Furthermore, only 0.46% (95%CI, −0.06–0.97%) of pregnant women reported neonatal death.

#### 3.2.3. Pre-Delta SARS-CoV-2 Infection during Pregnancy

Among pregnant women diagnosed during the pre-Delta period, 4.17% (95%CI, 1.53–6.80%) were admitted to ICU and 0.63% (95%CI, 0.05–1.20%) suffered maternal death; most patients had an asymptomatic infection (46.10%, 95%CI, 41.49–50.71%) and mild or moderate disease (54.4%, 95%CI, 28.06–80.74%), 10.74% (95%CI, 6.05–15.46%) needed respiratory support, and 15.81% (95%CI, 12.44–19.19%) received pharmacological treatment. In addition, the PP for eclampsia or preeclampsia (PP = 23.00%, 95%CI, 16.96–29.03%) was higher, while the PP for stillbirths (PP = 3.21%, 95%CI, −2.18–8.60%) was lower compared with those infected with the Delta and Omicron variants.

#### 3.2.4. SARS-CoV-2 Delta Variant Infection during Pregnancy

Among pregnant women infected with the Delta variant, 11.31% (95%CI, 4.00–18.61%) were admitted to the intensive care unit (ICU) and 4.20% (95%CI, 1.43–6.97%) suffered a maternal death; most patients had a mild or moderate illness (PP = 53.20%, 95%CI, 47.06–59.34%) and severe or critical illness (PP = 24.96%, 95%CI, 15.96–33.96%), 27.24% (95%CI, 20.51–33.97%) or required respiratory support, while 34.34% (95%CI, 28.17–40.42%) received pharmacological treatment, which was more severe than those diagnosed during the pre-Delta period or infected with Omicron. Among perinatal outcomes, the PPs of live births, stillbirths, and NICU were 83.62% (95%CI, 53.29–113.95%), 1.88% (95%CI, 0.45–3.31%), and 29.03% (95%CI, 8.02–50.04%), respectively. In addition, 33.85% (95%CI, 21.54–46.17%) reported preterm birth (<37 weeks), which was higher than those diagnosed during the pre-Delta period or for those infected with Omicron.

#### 3.2.5. SARS-CoV-2 Omicron Variant Infection during Pregnancy

Among pregnant women infected with the Omicron variant, 1.83% (95%CI, 0.85–2.81%) were admitted to the ICU and 0.40% (95%CI, −0.19–1.00%) had a maternal death, while most patients had an asymptomatic infection (PP = 53.00, 95%CI, 44.92–61.00%) and mild or moderate disease (PP = 46.00%, 95%CI, 38.61–53.38%). Only 1.11% (95%CI, 0.29–1.94%) had a severe or critical illness, and 2.63% (95%CI, 0.98–4.28%) required respiratory support. The PPs for preterm births (<37 weeks) and stillbirths were 10.63% (95%CI, −0.42–21.68%) and 1.13% (95%CI, −0.46%-2.72%), respectively, lower than those diagnosed during the pre-Delta period or infected with Delta.

### 3.3. Comparison of Maternal and Perinatal Outcomes of Delta and Pre-Delta SARS-CoV-2 Variant Infections during Pregnancy

Maternal and perinatal outcomes of Delta and pre-Delta SARS-CoV-2 variant infections during pregnancy were compared. The results showed that, compared with pre-Delta SARS-CoV-2 infection during pregnancy, there was a higher risk of adverse maternal outcomes and serious disease with a Delta infection; this included a higher risk of maternal ICU admission (OR = 2.68, 95%CI, 1.48–4.84), death (OR = 3.53, 95%CI, 2.72–4.60), and severe or critical illness (OR = 2.61, 95%CI, 1.70–4.02), and a lower risk of asymptomatic infection (OR = 0.27, 95%CI, 0.18–0.39). Compared with the pre-Delta infection group, pregnant women with Delta infection were more likely to receive pharmacological treatment (OR = 2.81, 95%CI, 1.92–4.10) and require respiratory support (OR = 2.92, 95%CI, 2.07–4.12), including nasal O_2_ support (intubation), non-invasive mechanical ventilation, invasive ventilation or ECMO, and ECMO (*p*-value < 0.05). In addition, results showed that Delta infection during pregnancy had a higher risk of preterm birth (<37 weeks) than pre-Delta infection (OR = 3.45, 95%CI, 1.17–10.15). No differences in other perinatal outcomes, pregnancy outcomes, complications, and forms of delivery were observed between the two groups. The results of the analysis are shown in Table 2 and Figure 2**.** These results suggest that the consequences of infection with the SARS-CoV-2 Delta variant during pregnancy might be a little more severe than those of pre-Delta infections.

### 3.4. Comparison of Maternal and Perinatal Outcomes of Omicron and Delta SARS-CoV-2 Variant Infections during Pregnancy

The analytical results suggest that the outcomes of maternal Omicron SARS-CoV-2 variant infection might be less severe than those of Delta variant infections. Compared with the Delta group, pregnant women with an Omicron infection were more likely to be asymptomatic (OR = 3.88, 95%, 2.42–6.23), and there might also be a lower risk of pregnant women with Omicron infection to suffer adverse maternal and perinatal outcomes such as maternal ICU admission (OR = 0.15, 95%, 0.024–0.97), death (OR = 0.22, 95%, 0.061–0.82), severe or critical illness (OR = 0.082, 95%, 0.040–0.17), and preterm birth <37 weeks (OR = 0.21, 95%, 0.11–0.40). In addition, Omicron infection during pregnancy was less likely to require respiratory support (OR = 0.063, 95%CI, 0.024–0.16), including nasal O_2_ support (intubation), non-invasive mechanical ventilation, invasive ventilation or ECMO, and ECMO, than Delta infection (*p*-value < 0.05). More results are shown in Table 3 and Figure 3**.**

### 3.5. Quality Evaluation and Publication Bias

The quality of the included cohort studies was evaluated based on the Newcastle–Ottawa quality assessment scale, and all were of good quality and had a low risk of bias (≥7 stars). Publication bias was evaluated for the comparison of maternal and perinatal outcomes among different variant infections during pregnancy based on Harbord’s modified test. Publication bias was suggested in ICU admission in the comparison between pre-Delta SARS-CoV-2 variants and Delta infection (*p* = 0.009), as well as in severe or critical illness in the comparison between Omicron and Delta infections (*p* = 0.042). The *p*-values for the modified Harbord’s test for other maternal and perinatal outcomes were higher than 0.1, indicating there was no publication bias.

## 4. Discussion

The emergence of variants of SARS-CoV-2 has added many challenges to the control and management of COVID-19. Pregnant women and newborns are at greater risk of SARS-CoV-2 infection and COVID-19-related severe illness due to their weakened immune functions. Understanding the potential risk of infection with different variants during pregnancy can better protect both maternal and infant health.

There is an increasing amount of data on maternal and infant outcomes after COVID-19 infection, though most describe outcomes after infection with only one variant. Our systematic review and meta-analysis of 18 articles, involving 133,058 cases infected with different SARS-CoV-2 variants during pregnancy, provided PP for 31 consequences, including maternal outcomes, disease severity, requirements for respiratory support and drug treatment, pregnancy outcomes and complications, forms of delivery, and perinatal outcomes. OR was also provided for different outcomes between Delta and pre-Delta periods, and between Omicron and Delta periods. This is the first systematic review and meta-analysis to comprehensively assess the association of different SARS-CoV-2 variant infections during pregnancy with maternal and perinatal outcomes.

The available data show that infection with different SARS-CoV-2 variants may have different effects on maternal and birth outcomes, and that Omicron infection was associated with less severe adverse maternal and neonatal issues, which was consistent with previous findings. A literature review showed that the number of cases and the proportion of ICU admissions in pregnant women increased significantly with Delta infection [34]. Pregnant women infected during the Delta period were more likely to have severe maternal outcomes, severe illness, or maternal death than those infected during the other variant periods [14,32,35]. A retrospective cohort study conducted in the United States found that, compared with the pre-Delta group, patients in the Delta variant group were more likely to require oxygen supplementation and were more likely to suffer from a serious illness [10]. The severity of disease in pregnant women infected during the Omicron period decreased. Emily H. Adhikari et al. found that the Delta period was associated with an increased incidence, while the Omicron period was associated with a decreased incidence of severe or critical illness compared with the pre-Delta period [12]. Evidence from a retrospective cohort study showed higher rates of maternal mortality among pregnant women who were unvaccinated, as well as the need for respiratory support during the Delta period, compared with those infected during the pre-Delta period. Conversely, similar levels were observed for those infected during the Omicron period [19]. Leonard Mndala et al. found that the incidences of stillbirths and neonatal deaths were lower in the Omicron period than in the Beta and Delta periods [14].

At present, the mechanism for pregnancy and birth outcomes related to COVID-19 infections is still unclear. Previous studies found that SARS-CoV-2 infections activated inflammatory cytokines causing severe placental damage, which often led to neonatal illness and death [36,37]. Another study found that fever and shortness of breath in pregnant women with COVID-19 increased the risk of maternal and neonatal complications [38]. However, the evidence is still limited. Therefore, it is hoped that more research will focus on the mechanism of adverse pregnancy and birth outcomes caused by COVID-19 infections in the future.

The association between Delta variant infection and more severe maternal and perinatal outcomes may result from a combination of maternal immunosuppression, high variant viral load, and the enhanced transmissibility of the Delta variant. On the one hand, various components of the immune system are weakened during pregnancy, leading to increased susceptibility to SARS-CoV-2 and decreased efficiency of viral clearance, and also to increased risk of excessive inflammation when faced with a SARS-CoV-2 infection [34]. On the other hand, the Delta variant has a higher viral load than the historical variant (20A.EU2), Beta, and Omicron in upper respiratory samples from infected cases; this could be one of the possible reasons for the greater transmissibility of the Delta variant [39,40]. Although Omicron is more prone to immune escape, its replication rate in the upper and lower respiratory tracts is significantly reduced [41]. In vitro experiments have also shown that Omicron has lower replication and fusion activity than Delta [42]. These may partly explain the milder infection of Omicron in pregnant women. In addition, more severe maternal and perinatal outcomes were observed in pregnant women who were not vaccinated [10,19,32,43]. The result of a prospective study showed that, even though none of the infants born to pregnant women infected with SARS-CoV-2 developed a serious illness, 18 out of 20 of the infants with non-severe illness came from cases who were eligible for vaccination but did not receive it [12]. Data from the meta-analysis showed that vaccine acceptance among pregnant women was unsatisfactory. Vaccine hesitancy and low vaccine acceptance can lead to the delayed establishment of an immune barrier, thus increasing maternal and perinatal health risks [44,45].

No subgroup analysis of vaccination rates was performed due to limited data, but evidence suggests that COVID-19 vaccination is effective in preventing infection and significantly reducing hospitalization and mortality rates in the general population [46], as well as in pregnant women. A meta-analysis showed that the risk of infection and hospitalization in pregnant women after receiving the COVID-19 vaccine was only half that of unvaccinated pregnant women, with no significant differences in adverse pregnant, fetal, and neonatal outcomes observed between the two groups [47]. Previous studies also found that among pregnant women infected with the Delta variant, the unvaccinated are more likely than the vaccinated to develop severe illness, laboratory or imaging abnormalities, and a greater need for respiratory support [10,48]. In addition, evidence has shown that vaccination or natural infection with SARS-CoV-2 during pregnancy can induce effective neutralizing antibodies (NAb) within maternal and cord blood [49], suggesting that antibodies produced after vaccination during pregnancy can be transferred to the fetus through the placenta, thereby establishing an immune barrier for the fetus against COVID-19.

Antibody levels decrease over time, meaning that full vaccination and a booster could increase antibody levels. Benjamin L. Sievers et al. found that neutralization titers against different SARS-CoV-2 variants increased after the second dose, and were 128, 91, 40, and 10.2 times higher for wild type, Delta, Beta, and Omicron, respectively, than at baseline [50]. Chingju Shen et al. found that, among pregnant women who were vaccinated with the Moderna vaccine, antibody levels in both maternal serum and cord blood were significantly higher in the fully vaccinated group than in the single-dose group [51]. Ellen Øen Carlsen et al. found that among infants born to women who received COVID-19 vaccines, the risk of a neonatal positive SARS-CoV-2 test was lower in the vaccinated group than in the unvaccinated group, and lower in those who had received a third dose than those with only two doses [52]. In conclusion, vaccination against COVID-19 is associated with reduced disease severity and improved outcomes in pregnant women, and the importance of increasing vaccination coverage among pregnant women should be considered. As COVID-19 remains a global pandemic, and new variants continue to emerge, it is recommended that research on the impact of new variants on maternal and infant health should be strengthened in the future.

There are certain limitations to this study. Firstly, due to limited data, only a few outcomes were analyzed for certain variants. Secondly, no subgroup analysis was conducted for limited data. Nonetheless, in the discussion section, a number of key factors were discussed, including the question of whether pregnant women should be vaccinated or not. Thirdly, publication bias existed in some of the results.

## 5. Conclusions

The results suggest that Omicron infection is associated with less severe maternal and neonatal adverse outcomes. Nonetheless, maternal ICU admission, the need for respiratory support, and preterm birth did still occur. Since Omicron is currently the predominant global strain, with the highest rate of transmission, it is still important to pay more attention to the vulnerable population of mothers and infants. In particular, obstetricians and gynecologists should not ignore the adverse risks of maternal ICU admission, respiratory support, and preterm birth in pregnant patients with SARS-CoV-2 infection, in order to protect the health of both mothers and infants.

## Figures and Tables

**Figure 1 ijerph-19-15932-f001:**
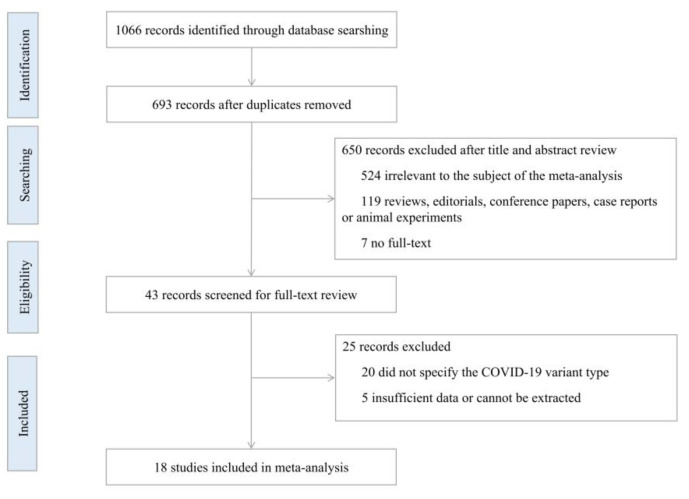
Flowchart for study selection.

**Figure 2 ijerph-19-15932-f002:**
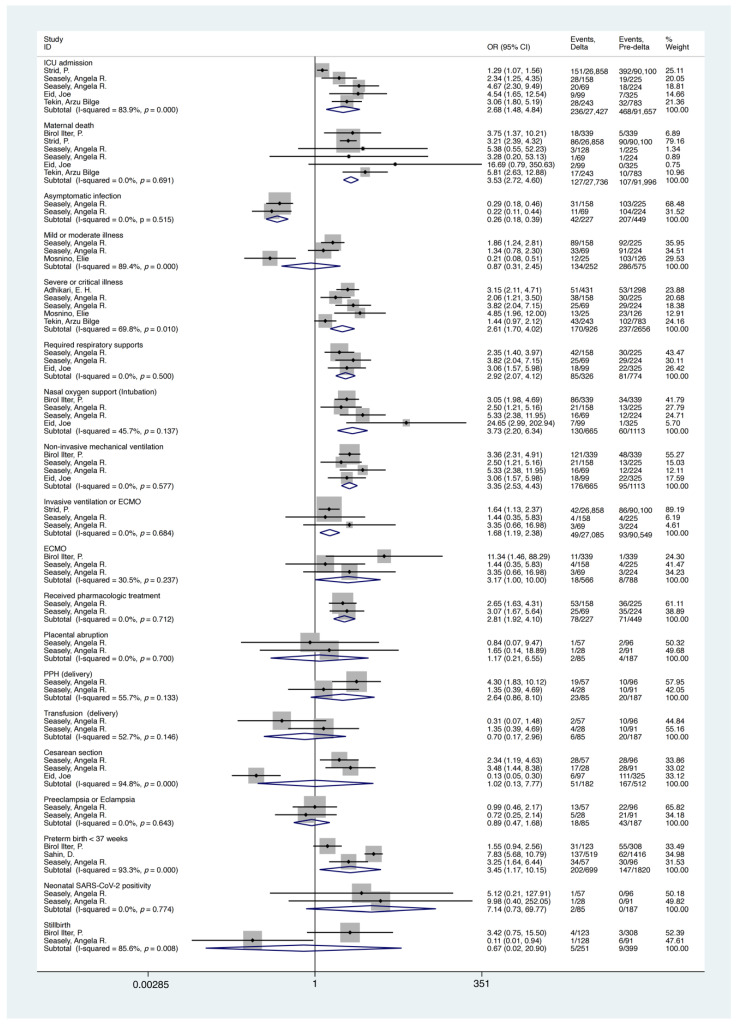
Forest plot of maternal and perinatal outcomes of Delta or pre-Delta SARS-CoV-2 variant infections during pregnancy. NOTE: Weights are from random effects analysis.

**Figure 3 ijerph-19-15932-f003:**
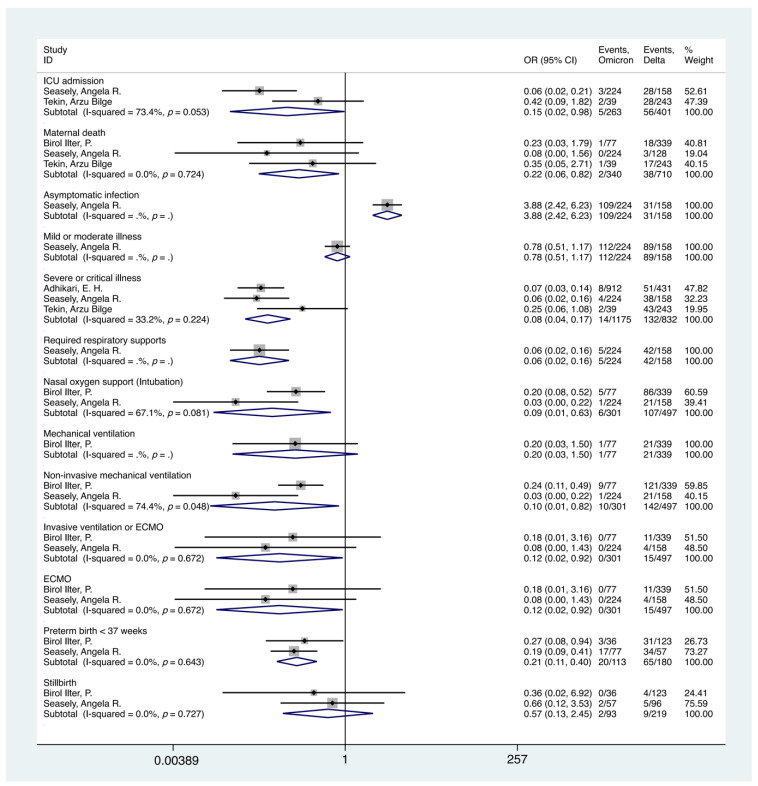
Forest plot of maternal and perinatal outcomes of Omicron or Delta SARS-CoV-2 variant infections during pregnancy. NOTE: Weights are from random effects analysis.

**Table 1 ijerph-19-15932-t001:** Pooled prevalence (PP) of maternal and perinatal outcomes of different SARS-CoV-2 variant infections during pregnancy.

Outcomes	Type of SARS-CoV-2 Variant					
	Wild Type (%)	Alpha (%)	Alpha and Gamma (%)	Pre-Delta (%)	Delta (%)	Omicron (%)
Maternal Outcomes						
ICU admission	-	-	-	4.17 (1.53–6.80) *	11.31 (4.00–18.61) *	1.83 (0.85–2.81) *
Maternal death	0.39 (−0.44–1.22)	0.28 (0.04–0.53) *	-	0.63 (0.05–1.20) *	4.20 (1.43–6.97) *	0.40 (−0.19–1.00)
Severity of Disease						
Asymptomatic infection	-	-	-	46.10 (41.49–50.71) *	18.73 (13.34–23.40) *	53.00 (44.92–61.00) *
Mild or moderate illness	-	-	-	54.40 (28.06–80.74) *	53.20 (47.06–59.34) *	46.00 (38.61–53.38) *
Severe or critical illness	-	-	-	11.99 (6.23–17.74) *	24.96 (15.96–33.96) *	1.11 (0.29–1.94) *
Form of Respiratory Support						
Required respiratory support	13.40 (10.55–16.25) *	-	-	10.74 (6.05–15.46) *	27.24 (20.51–33.97) *	2.63 (0.98–4.28) *
Nasal O_2_ support (intubation)	-	-	-	5.23 (0.44–10.02) *	16.24 (8.56–23.91) *	1.47 (−0.52–3.47)
Mechanical ventilation	-	-	-	-	4.02 (−0.44–8.48)	1.30 (−1.23–3.83)
Non-invasive mechanical ventilation	7.45 (1.01–13.89) *	14.79 (3.45–26.13) *	-	7.87 (4.32–11.41) *	22.55 (15.60–29.51) *	3.87 (0.33–7.41) *
Invasive ventilation or ECMO	1.83 (−0.15–3.82)	4.21 (2.82–5.61) *	7.35 (1.15–13.56) *	0.83 (−0.34–1.99)	1.79 (0.22–3.37) *	0.74 (−0.71–2.19)
ECMO	0.16 (−0.74–1.06)	0.54 (0.03–1.06) *	7.35 (1.15–13.56) *	0.87 (−0.08–1.82)	3.10 (1.67–4.53) *	-
Drug Therapy						
Received pharmacological treatment	-	-	-	15.81 (12.44–19.19) *	34.34 (28.17–40.42) *	-
Pregnancy Outcome						
Intrauterine fetal demise	-	-	-	-	1.82 (0.77–2.88) *	-
Placental abruption	-	-	-	2.14 (0.07–4.21) *	1.60 (0.60–2.60) *	-
Transfusion (delivery)	-	-	-	10.69 (6.26–15.12) *	7.14 (−2.84–17.13)	-
PPH (delivery)	-	-	-	10.69 (6.26–15.12) *	23.93 (5.27–42.60) *	-
Abortion	2.65 (2.05–3.26) *	-	-	-	2.91 (−0.66–6.49)	-
Form of Delivery						
Spontaneous vaginal delivery	47.00 (27.20–66.80) *	50.18 (20.16–80.19) *	-	-	23.89 (12.31–35.48) *	-
Cesarean section	41.70 (31.40–52.00) *	37.32 (24.58–50.05) *	-	32.54 (28.49–36.60) *	29.35 (16.92–41.79) *	32.96 (28.07–37.85) *
Pregnancy Complication						
Preeclampsia or Eclampsia	1.06 (0.53–1.59) *	-	-	23.00 (16.96–29.03) *	9.63 (1.47–17.80) *	12.52 (−10.30–35.33)
Gestational diabetes	-	-	-	-	2.58 (1.38–3.77) *	-
Gestational hypertension	6.62 (−4.84–18.08)	-	-	-	6.34 (0.33–12.34) *	-
Deep vein thrombosis or pulmonary embolism	0.21 (−0.03–0.45)	-	-	-	-	-
Pneumonia	18.95 (17.01–20.90) *	-	-	-	-	-
Perinatal Outcome						
Preterm birth < 37 weeks	-	36.12 (5.37–66.87) *	-	18.58 (9.52–27.65) *	33.85 (21.54–46.17) *	10.63 (−0.42–21.68)
Preterm birth < 34 weeks	-	-	-	-	-	2.78 (−2.59–8.15)
Live birth	81.70 (44.94–118.45) *	-	-	-	83.62 (53.29–113.95) *	-
Stillbirth	0.89 (0.56–1.22) *	0.77 (0.16–1.38) *	-	3.21 (−2.18–8.60)	1.88 (0.45–3.31) *	1.13 (−0.46–2.72)
NICU admission	12.20 (7.98–16.41) *	16.85 (9.05–24.66) *	-	-	29.03 (8.02–50.04) *	-
Neonatal SARS-CoV-2 positivity	-	-	-	-	2.11 (−0.94–5.17)	-
Neonatal death	0.29 (−0.04–0.62)	-	0.46 (−0.06–0.97)	-	-	-

* *p*-value < 0.05.

**Table 2 ijerph-19-15932-t002:** Maternal and perinatal outcomes of Delta or pre-Delta SARS-CoV-2 variant infections during pregnancy.

Outcomes	Data Source	Delta Variant (*n*/N)	Pre-Delta (*n*/N)	OR	95%CI	*p*-Value	I^2^%	*p*-Heterogeneity
Maternal Outcomes								
ICU admission	2, 8, 9, 11, 16	236/27,427	468/91,657	2.679	1.482–4.841	<0.05	83.9	<0.05
Maternal death	1, 2, 8, 9, 11, 16	127/27,736	107/91,996	3.533	2.715–4.597	<0.05	0	>0.05
Severity of Disease								
Asymptomatic infection	8, 9	42/227	20/449	0.265	0.179–0.391	<0.05	0	>0.05
Mild or moderate illness	8, 9, 10	134/252	286/575	0.868	0.308–2.449	>0.05	89.4	<0.05
Severe or critical illness	3, 8, 9, 10, 16	179/926	237/2656	2.614	1.701–4.018	<0.05	69.7	<0.05
Form of Respiratory Support								
Required respiratory support	8, 9, 11	85/326	81/774	2.919	2.069–4.118	<0.05	0	>0.05
Nasal O_2_ support (intubation)	1, 8, 9, 11	130/665	60/1113	3.732	2.196–6.343	<0.05	45.7	>0.05
Non-invasive mechanical ventilation	1, 8, 9, 11	176/665	95/1113	3.346	2.527–4.431	<0.05	0	>0.05
Invasive ventilation or ECMO	2, 8, 9	49/27,085	93/90,549	1.68	1.186–2.381	<0.05	0	>0.05
ECMO	1, 8, 9	18/566	8/788	3.133	1.033–9.501	<0.05	25.7	>0.05
Drug Therapy								
Received pharmacological treatment	8, 9	78/227	71/449	2.805	1.919–4.102	<0.05	0	>0.05
Pregnancy Outcome								
Placental abruption	8, 9	2/85	4/187	1.174	0.210–6.547	>0.05	0	>0.05
PPH (delivery)	8, 9	23/85	20/187	2.642	0.861–8.104	>0.05	55.7	>0.05
Transfusion (delivery)	8, 9	6/85	20/187	0.702	0.169–2.919	>0.05	51.7	>0.05
Form of Delivery								
Cesarean section	8, 9, 11	51/182	167/512	1.017	0.133–7.767	>0.05	94.8	<0.05
Pregnancy Complication								
Preeclampsia or Eclampsia	8, 9	18/85	43/187	0.892	0.474–1.680	>0.05	0	>0.05
Perinatal Outcomes								
Preterm birth < 37 weeks	1, 4, 8	202/699	147/1820	3.451	1.173–10.148	<0.05	93.3	<0.05
Neonatal SARS-CoV-2 positivity	8, 9	2/85	0/187	7.143	0.731–69.773	>0.05	0	>0.05
Stillbirth	1, 9	5/251	9/399	0.67	0.021–20.900	>0.05	85.6	<0.05

**Table 3 ijerph-19-15932-t003:** Maternal and perinatal outcomes of Omicron or Delta SARS-CoV-2 variant infections during pregnancy.

Outcomes	Data Source	Omicron Variant (*n*/N)	Delta Variant (*n*/N)	OR	95%CI	*p*-value	I^2^%	*p*-Heterogeneity
Maternal Outcomes								
ICU admission	8, 16	5/263	56/401	0.154	0.024–0.974	<0.05	73.3	>0.05
Maternal death	1, 8, 16	2/340	38/710	0.224	0.061–0.820	<0.05	0	>0.05
Severity of Disease								
Asymptomatic infection	8	109/224	31/158	3.883	2.422–6.225	<0.05	-	-
Mild or moderate illness	8	112/224	89/158	0.775	0.515–1.167	>0.05	-	-
Severe or critical illness	3, 8, 16	14/1175	132/832	0.082	0.040–0.171	<0.05	32.9	>0.05
Form of Respiratory Support								
Required respiratory support	8	5/224	42/158	0.063	0.024–0.164	<0.05	-	-
Nasal O_2_ support (intubation)	1, 8	6/301	107/497	0.096	0.015–0.614	<0.05	65.9	>0.05
Mechanical ventilation	1	1/77	21/339	0.199	0.016–1.504	>0.05	-	-
Non-invasivemechanical ventilation	1, 8	10/301	14/497	0.104	0.014–0.777	<0.05	72.8	>0.05
Invasive ventilation or ECMO	1, 8	0/301	15/497	0.12	0.016–0.925	<0.05	0	>0.05
ECMO	1, 8	0/301	15/497	0.12	0.016–0.925	<0.05	0	>0.05
Perinatal Outcomes								
Preterm birth < 37 weeks	1, 8	20/113	65/180	0.21	0.110–0.401	<0.05	0	>0.05
Stillbirth	1, 7	2/93	9/219	0.572	0.133–2.450	>0.05	0	>0.05

## Data Availability

Data are available from the corresponding author by request.

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
