# Peer review of "Association of Infection with Different SARS-CoV-2 Variants during Pregnancy with Maternal and Perinatal Outcomes: A Systematic Review and Meta-Analysis"

_ijerph, 2022, doi:10.3390/ijerph192315932_

Round 1

Reviewer 1 Report

Association of different SARS-CoV-2 variants infection during 2 pregnancy with maternal and perinatal outcomes: a systematic 3 review and meta-analysis

The article review evidence for differences in pregnancy and birth outcomes according to variant of maternal COVID-infection. The literature review seems to be carried out in a sound way, and the results presented in a reasonable way. One general comment for the discussion (also in the abstract)is more practical implications of the research, as there is not likely that pregnant women are checked for type of COVID-infection and treated thereafter, or that the variants investigated will be the ones frequently seen in the time to come. What should we learn for future pandemics?

Abstract

What do you mean by “in the real world”? Or what would the opposite be?

The last sentence is extremely long, suggest to divide into shorter sentences, easier to read.

Methods

Exclusion criteria (1) irrelevant to the subject of the meta-analysis, such as studies that did not use SARS-CoV-2 vaccination as the exposure: nothing builds up to excluding articles not using vaccination as exposure? All the introduction is about infections.

Results

Table 1 is very difficult to read and is in need of better layout

Table 2: which differences are p-values pointing to?

Discussion

See general comment above

It would have been nice with an interpretation of how important COVID-19 infections is for maternal and birth outcomes, and how important the variation between VOCs are in this.

It is also not clear to me whether adverse pregnancy and birth outcomes related to COVID-19 infections are due to the virus type per se, or to having an adverse course of disease from the corona virus, or both?

Author Response

The article review evidence for differences in pregnancy and birth outcomes according to variant of maternal COVID-infection. The literature review seems to be carried out in a sound way, and the results presented in a reasonable way. One general comment for the discussion (also in the abstract) is more practical implications of the research, as there is not likely that pregnant women are checked for type of COVID-infection and treated thereafter, or that the variants investigated will be the ones frequently seen in the time to come. What should we learn for future pandemics?

Response: Thanks for the reviewer’s suggestion. We have added the general comment for the discussion: Omicron infection was associated with less severe maternal and neonatal adverse outcomes, however, maternal ICU admission, need for respiratory support, and preterm birth did occur. Since Omicron is currently the predominant strain globally, and has the highest transmission ability, it is still important to pay more attention to the protection of the vulnerable population of mothers and infants. In particular, obstetricians and gynecologists should not ignore the adverse risks of maternal ICU admission, respiratory support, and preterm birth in pregnant patients with SARS-CoV-2 infection to protect the health of mothers and infants. As COVID-19 remains a pandemic and new variants continue to emerge, we recommend that research on the impact of new variants on maternal and infant health should be strengthened in the future.

Abstract

What do you mean by “in the real world”? Or what would the opposite be?

Response: Thanks for the reviewer’s suggestion. Real-world study (RWS) and traditional RCTs are both commonly used study designs in evidence-based medicine, and their differences are in the environment in which the data are generated. The data of traditional RCTs are generated from clinical settings and carried out under relatively strict conditions. RWS is a research based on the actual diagnosis and treatment and daily life scenarios, using the health-related information recorded by the data platform. (Reference: Blonde L, Khunti K, Harris SB, Meizinger C, Skolnik NS. Interpretation and Impact of Real-World Clinical Data for the Practicing Clinician. Adv Ther. 2018;35(11):1763-1774. doi:10.1007/s12325-018-0805-y)

The last sentence is extremely long, suggest to divide into shorter sentences, easier to read.

Response: Thanks for the reviewer’s suggestion. We have divided the last sentence into shorter sentences in the abstract section.

Methods

Exclusion criteria (1) irrelevant to the subject of the meta-analysis, such as studies that did not use SARS-CoV-2 vaccination as the exposure: nothing builds up to excluding articles not using vaccination as exposure? All the introduction is about infections.

Response: Thanks for the reviewer’s suggestion. We have revised the description of exclusion criteria: (1) irrelevant to the subject of the meta-analysis, such as studies that did not use SARS-CoV-2 infections during pregnancy as the exposure.

Results

Table 1 is very difficult to read and is in need of better layout

Response: Thanks for the reviewer’s suggestion. Table 1 shows the basic characteristics of the included studies. We have uploaded it as supplementary materials.

Table 2: which differences are p-values pointing to?

Response: Thanks for the reviewer’s suggestion. We used Stata version 16.0 (Stata Corp, College Station, Texas, USA) to pool the prevalence of maternal and perinatal outcomes of different SARS-CoV-2 variants infection during pregnancy and reported their confidence intervals (CI), as shown in Table 1 (original Table 2), p < 0.05 was considered to be statistic significance, otherwise it was not statistic significance.

Discussion

See general comment above

It would have been nice with an interpretation of how important COVID-19 infections is for maternal and birth outcomes, and how important the variation between VOCs are in this.

Response: Thanks for the reviewer’s suggestion. We have added an interpretation in the discussion: Available data showed that infection with different SARS-CoV-2 variants may have different effects for maternal and birth outcomes, and there was a higher risk of adverse maternal outcomes and serious disease with Delta infection compared with SARS-CoV-2 pre-Delta and Omicron infection during pregnancy.

It is also not clear to me whether adverse pregnancy and birth outcomes related to COVID-19 infections are due to the virus type per se, or to having an adverse course of disease from the corona virus, or both?

Response: Thanks for the reviewer’s suggestion. At present, the mechanism of pregnancy and birth outcomes related to COVID-19 infections is still unclear. Previous studies found that SARS-CoV-2 infection activated inflammatory cytokines and caused severe placental damage, which often led to neonatal illness and death. [1,2] Another study found that fever and shortness of breath in pregnant women with COVID-19 increased the risk of maternal and neonatal complications. [3] However, the evidence is still limited. Therefore, we hope that more research will focus on the mechanism of adverse pregnancy and birth outcomes caused by COVID-19 infections in the future. We have added this point in the discussion.

References:

[1] Boelig RC, Aghai ZH, Chaudhury S, et al. Impact of COVID-19 disease and COVID-19 vaccination on maternal or fetal inflammatory response, placental pathology, and perinatal outcomes. Am J Obstet Gynecol. 2022;227(4):652-656. doi:10.1016/j.ajog.2022.05.049

[2] Huynh A, Sehn JK, Goldfarb IT, et al. SARS-CoV-2 Placentitis and Intraparenchymal Thrombohematomas Among COVID-19 Infections in Pregnancy. JAMA Netw Open. 2022;5(3):e225345. doi:10.1001/jamanetworkopen.2022.5345

[3] Villar J, Ariff S, Gunier RB, et al. Maternal and Neonatal Morbidity and Mortality Among Pregnant Women With and Without COVID-19 Infection: The INTERCOVID Multinational Cohort Study. JAMA Pediatr.2021;175(8):817–826. doi:10.1001/jamapediatrics.2021.1050

Reviewer 2 Report

This topic is of great relevance and is timely, given that while the severity of the pandemic has decreased, there are still current and new variants, waves of illness persist, and COVID-19 is likely to remain in some form. The data is presented well, however the discussions are so lengthy in sections 3.3 and 3.5, it is suggested to include some sub-titles to divide the text for purposes of information organization. Otherwise, very well done! 

Author Response

This topic is of great relevance and is timely, given that while the severity of the pandemic has decreased, there are still current and new variants, waves of illness persist, and COVID-19 is likely to remain in some form. The data is presented well, however the discussions are so lengthy in sections 3.3 and 3.5, it is suggested to include some sub-titles to divide the text for purposes of information organization. Otherwise, very well done!

Response: Thanks for the reviewer’s suggestion. We have added some sub-titles to divide the text in section 3.2 into several discussion parts, seen in pages 6-7: 3.2.1. Wild type SARS-CoV-2 infection during pregnancy, 3.2.2. SARS-CoV-2 Alpha or Gamma variants infection during pregnancy, 3.2.3. Pre-Delta SARS-CoV-2 infection during pregnancy, 3.2.4. SARS-CoV-2 Delta variant infection during pregnancy, 3.2.5. SARS-CoV-2 Omicron variant infection during pregnancy.

Reviewer 3 Report

I congratulate you on your interesting article titled ‘Association of different SARS-CoV-2 variants infection during pregnancy with maternal and perinatal outcomes: a systematic review and meta-analysis’

There are a few observations.

The article appears to have been submitted in a hurry as there are quite a lot of grammatical errors which need to be corrected. There are inappropriate use of tenses, for example page 3, line 112 reads ‘will be excluded’ instead of ‘were excluded’ There are a lot more of such errors and the authors need to properly edit the article before resubmission.

Inclusion criteria have not been well defined.

Figure 1: flow chart of study selection suggests that 7 studies were excluded as there were no full texts available. This appears to contradict the statement on page 4, line 161 which states that ‘Among the 50 studies under full text review…’

Part of the exclusion criteria was ‘studies that did not use SARS-CoV2 vaccination as the exposure (page 3, line 112)’, it would appear that some of the studies included fell under this category.

Results and discussion can be presented in a clearer way.

Some references in the text are cited with the first names of the author, instead of the surname.

Author Response

The article appears to have been submitted in a hurry as there are quite a lot of grammatical errors which need to be corrected. There are inappropriate use of tenses, for example page 3, line 112 reads ‘will be excluded’ instead of ‘were excluded’ There are a lot more of such errors and the authors need to properly edit the article before resubmission.

Response: Thanks for the reviewer’s suggestion. We have reviewed the full text carefully and revised the grammatical errors.

Inclusion criteria have not been well defined.

Response: Thanks for the reviewer’s suggestion. We have refined the definition of inclusion criteria, seen in page 3, lines 107-110: (1) studies explored maternal and perinatal outcomes of SARS-CoV-2 infection during pregnancy, (2) specified the type of maternal SARS-CoV-2 variant infection or the dominant epidemic strains while infection, and (3) published studies or preprint studies.

Figure 1: flow chart of study selection suggests that 7 studies were excluded as there were no full texts available. This appears to contradict the statement on page 4, line 161 which states that ‘Among the 50 studies under full text review…’

Response: Thanks for the reviewer’s suggestion. We have revised the flow chart and statement of study selection, see in Figure 1 and page 4, line 151: After reading the titles and abstracts, 650 articles were excluded based on the inclusion and exclusion criteria. Among the 43 studies under full-text review, 25 studies were excluded.

Part of the exclusion criteria was ‘studies that did not use SARS-CoV2 vaccination as the exposure (page 3, line 112)’, it would appear that some of the studies included fell under this category.

Response: Thanks for the reviewer’s suggestion. We have revised the description of exclusion criteria: (1) irrelevant to the subject of the meta-analysis, such as studies that did not use SARS-CoV-2 infections during pregnancy as the exposure.

Results and discussion can be presented in a clearer way.

Response: Thanks for the reviewer’s suggestion. We have adjusted the statements of results and discussion parts.

Some references in the text are cited with the first names of the author, instead of the surname.

Response: Thanks for the reviewer’s suggestion. We have revised the citation of the authors' name.